# A general supramolecular strategy for fabricating full-color-tunable thermally activated delayed fluorescence materials

Nan Xue[1,3], He-Ye Zhou[2,3], Ying Han[2], Meng Li[1,2], Hai-Yan Lu ®[1] ✉ & Chuan-Feng Chen ®[1,2] ✉

Developing a facile and feasible strategy to fabricate thermally activated delayed fluorescence materials exhibiting full-color tunability remains an appealing yet challenging task. In this work, a general supramolecular strategy for fabricating thermally activated delayed fluorescence materials is proposed. Consequently, a series of host–guest cocrystals are prepared by electron-donating calix[3]acridan and various electron-withdrawing guests. Owing to the through-space charge transfer mediated by multiple noncovalent interactions, these cocrystals all display efficient thermally activated delayed fluorescence. Especially, by delicately modulating the electron-withdrawing ability of the guest molecules, the emission colors of these cocrystals can be continuously tuned from blue (440 nm) to red (610 nm). Meanwhile, high photoluminescence quantum yields of up to 87% is achieved. This research not only provides an alternative and general strategy for the fabrication of thermally activated delayed fluorescence materials, but also establishes a reliable supramolecular protocol toward the design of advanced luminescent materials.

Thermally activated delayed fluorescence (TADF) materials have received immense attention in recent years owing to their wide applications in various kinds of research fields including optoelectronic devices[1–3], photocatalysis[4,5], laser displays[6], luminescence imaging and sensing[7,8]. In terms of fundamental studies and practical applications, the design and fabrication of TADF materials exhibiting color-tunable emission properties is particularly important to achieve multi-color displays and to meet the need of next generation light-emitting materials[9–11]. In order to obtain TADF materials with different emission colors, it is the almost universally adopted to modify the chemical structure of luminophores by the covalent syntheses. Consequently, various examples of single-component color-tunable TADF materials featuring intramolecular charge-transfer (CT) excited states have been reported[12,13]. However, such systems inevitably suffer from complicated molecular designs and time-consuming chemical

synthesis[14,15]. Alternatively, the intermolecular donor–acceptor (D – A) systems seem to be a promising candidate to circumvent this problem, in which the emission colors can be readily tuned by simply altering and blending different donor and acceptor molecules[16–18]. Inter-molecular D – A type TADF materials, however, are often uncontrollable and unpredictable owing to the long-range intermolecular CT states that occurs with random frequency[19,20], making it difficult to customize emission colors at will, not to mention realizing full-color emissions and high photoluminescence quantum yields (PLQYs)[21]. Therefore, developing an easy-to-operate and effective method to finely modulate emission colors of TADF materials remains an appealing yet challenging task.

During the past few decades, the employment of supramolecular strategies, including host–guest complexation, molecular machine, supramolecular polymerization, supramolecular self-assembly and so

[1]University of Chinese Academy of Sciences, Beijing 100049, China. [2]Beijing National Laboratory for Molecular Sciences, CAS Key Laboratory of Molecular Recognition and Function, Institute of Chemistry, Chinese Academy of Sciences, Beijing 100190, China. [3]These authors contributed equally: Nan Xue, He-Ye Zhou. ✉e-mail: haiyanlu@ucas.ac.cn; cchen@iccas.ac.cn

on, has presented opportunities for the fabrication of organic luminescent materials and the regulation of their photophysical properties[22,23]. Distinguished from traditional covalent chemistry, the introduction of supramolecular strategies into luminescence systems has excellent advantages, which can not only considerably simplify the preparation of materials through the spontaneous assembly of molecular building modules, but can also facilely tune the luminescent properties by manipulating intermolecular noncovalent interactions. Based on supramolecular strategies, a large number of supramolecular luminescent materials have been successfully designed and constructed[24–34]. For example, by means of host–guest complexation, Tian and Ma's group realized multicolor fluorescence including white light based on γ-cyclodextrin[35], and Liu et al. fabricated supramolecular room-temperature phosphorescent materials with tunable emission based on cucurbit[n]urils[36,37]. However, the luminescent materials fabricated by supramolecular strategy are mainly focused on traditional fluorescent and phosphorescent properties[38]. In sharp contrast, the tuning of TADF property by supramolecular strategy is still far from being well-developed for the lack of reliable and versatile supramolecular platforms[39].

Previously, we[40] unexpectedly found that the host–guest CT complex formed by calix[3]acridan (C[3]A) and 1,2-dicyanobenzene could exhibit TADF emission, which offered a supramolecular tool to fabricate TADF materials (Fig. 1a). On this basis, we further speculated whether it was a general supramolecular strategy to fabricate various TADF materials based on the intermolecular charge-transfer (CT) interactions between the macrocyclic donor and various acceptors. Especially, following this supramolecular strategy, we wondered if the TADF materials with multi-color or even full-color emissions could be easily achieved only by modulating the electron-withdrawing ability of the guest molecules. Additionally, the efficiency extremes of luminescence from supramolecular TADF materials deserve to be explored.

In this work, we report a series of crystalline host–guest complexes formed by C[3]A and various electron-withdrawing guests. It is found that all these cocrystal materials exhibit excellent TADF properties, and their emission colors can be easily and continuously tuned from blue to red with the emission wavelengths changing over a 170 nm range. Furthermore, an exceptionally high PLQY of up to 87% for the supramolecular TADF materials can be achieved, which represents the highest value among the reported intermolecular D–A type TADF materials so far. This work manifests the great potential of such supramolecular strategy, paving a general and efficient approach toward the design and preparation of full-color-tunable TADF materials.

## Results

### Design concept of color-tunable supramolecular TADF materials

For efficient TADF emission, a very small energy gap ($\Delta E_{ST}$) between the lowest singlet ($S_1$) and triplet ($T_1$) excited states is required to facilitate the reverse intersystem crossing (RISC) process from $T_1$ to $S_1$. According to the quantum mechanical analysis, the reduction in the overlap of the highest occupied molecular orbital (HOMO) and the lowest unoccupied molecular orbital (LUMO) can realize small $\Delta E_{ST}$[41,42]. Following this principle, intramolecular and intermolecular D–A systems with pronounced CT character have been widely reported to have the ability to generate TADF emission. From supramolecular perspective, host–guest complexation can serve as a good platform to achieve through-space CT excited state between the host and guest[43,44], making it much easier to modulate the superstructures and properties of the materials by taking advantage of dynamic and reversible noncovalent interactions. So conceptually, it is possible to tune the emission color of TADF materials by the marriage of TADF mechanisms and host–guest complexation.

The macrocycle C[3]A, containing three electron-donating acridan subunits, can encapsulate electron-withdrawing molecules with appropriate shapes and sizes to form host–guest CT complexes in the solid state, to realize TADF emission that occurs as a result of electron transient from the LUMO of guest to the HOMO of C[3]A[40]. Hence, it is possible to tune the emission wavelength of the supramolecular TADF materials by regulating the LUMO of the guest/acceptor (Fig. 1b). As we all know, the cyano group has a strong electron-accepting ability to produce a significant increase in the electron affinity of the resulting cyano-substituted aromatics, which has been widely employed to construct D–A type TADF materials[45,46]. Accordingly, we selected a series of cyano-substituted aromatic acceptors (Fig. 1c), 2-cyanopyridine (G1), 1,3-dicyanobenzene (G2), 3,5-dicyanofluorobenzene (G3), 3,5-dicyanopyridine (G4), 1,3,5-tricyanobenzene (G5), 2,4-dicyanopyridine (G6) and 3,4-dicyanopyridine (G7), as guest molecules to fabricate TADF materials through the formation of host–guest cocrystals. Among these guests, the number and position of the cyano group are carefully adjusted to change their electron-withdrawing ability. Firstly, the electrostatic potential (ESP) maps of C[3]A and G1-G7 were calculated to evaluate the possibility of the formation of host–guest inclusion complexes. It was found that the acridan panels of C[3]A were highly electronegative. Conversely, the aromatic areas of G1-G7 displayed strong electropositivity, implying an underlying trend to form CT interactions between C[3]A and G1-G7. Subsequently, we investigated the energy levels of G1-G7 by quantum

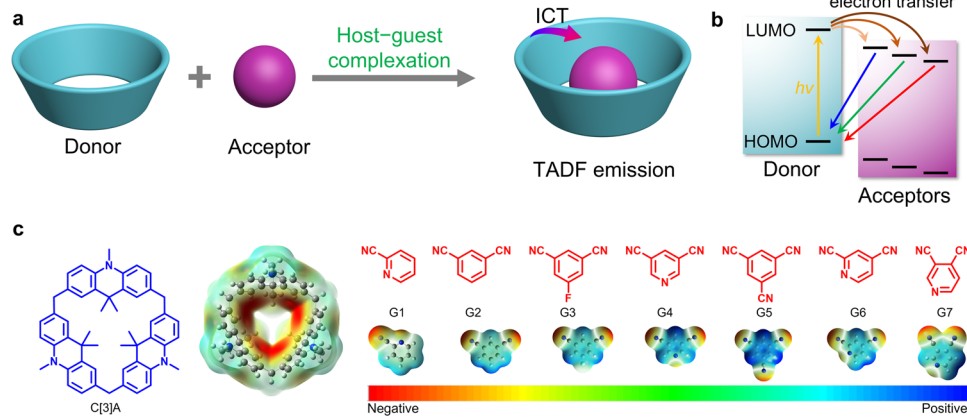

**Fig. 1 | Design concept of color-tunable supramolecular TADF materials. a** The realization of TADF emission by formation of intermolecular charge-transfer excited state between the macrocyclic donor and guest. **b** Scheme of color-tunable TADF emission formed between macrocyclic donor and guests with HOMO and LUMO levels, respectively. **c** Chemical structures and electrostatic potential maps of calix[3]acridan (C[3]A) and electron-withdrawing guests (G1-G7).

chemical calculations and found that the LUMO levels of these guests were lowered from −1.83 to −2.91 eV in order of G1 - G7 (Supplementary Table 1), indicating their gradually enhanced electron-withdrawing ability.

## Preparation of host-guest cocrystals

In general, the preparation of organic cocrystals composed of two or more different constituent molecules is not trivial[47]. It is difficult to make sure that the designed molecules can produce desirable cocrystals, which hinder the widespread use of cocrystal engineering toward functional materials. In this regard, supramolecular-macrocycle-based co-crystallization method seems to settle this problem on account of their diverse complexation properties and adaptive binding abilities[48], which is well demonstrated in this work. By employing solvent evaporation or diffusion of the mixture solution of the 1:1 C[3]A and guests, a series of high-quality host–guest cocrystals, namely G1@C[3]A - G7@C[3]A, with a blocky morphology with lengths of up to a few hundreds of micrometers were obtained in a facile way (Supplementary Figs. 4–10). The crystallographic data for all host–guest cocrystals were summarized in Supplementary Tables 2–8. The formation of these cocrystals can be observed with the naked eye on account of their dramatic colorimetric and fluorescence change (Fig. 2). For example, the cocrystals G6@C[3]A and G7@C[3]A are yellow under sunlight, which are visually different from those for their individual components. More interestingly, the fluorescence microscopy images revealed that these cocrystals exhibit significant fluorescence changes under UV light in sharp contrast to their precursors that lack visible solid-state fluorescence, indicating the influence of cocrystallization upon the photophysical behavior. Moreover, their emission colors continuously change from blue to red along with varied guests, which can be attributed to varying degrees of the CT interactions between C[3]A and guests. These results pointed up the fact that a wide range of color-tuning of solid-state luminescence was successfully achieved in our systems.

## Structural analysis of host-guest cocrystals

To further confirm the chemical structures of as-prepared cocrystals, the single-crystal X-ray diffraction (SCXRD) analysis were performed. It is found that the G1@C[3]A and G5@C[3]A adopt monoclinic $P2_1/c$ and $P2_1/n$ space group, respectively. Differently, the G2@C[3]A, G3@C[3]A, G4@C[3]A, G6@C[3]A, G7@C[3]A all crystallize in the triclinic space group $P–1$. As shown in the crystal structures (Fig. 3a), C[3]A in these cocrystals exclusively adopts a cone conformation and forms well-defined inclusion complexes with G1 - G7. All the guests are bound inside the cavity of C[3]A from the upper rim with a 1:1 host–guest or stoichiometric ratio. In these crystal structures, C[3]A slightly adjusts its conformation through flipping the acridan panels in order to best fit

the guest molecules. Owing to this great degree of structural flexibility, C[3]A represents a more universal macrocyclic platform to construct host–guest crystalline materials compared with other known macrocyclic host molecules[49–52].

The internal driving force for these complexation processes in the solid state is the formation of multiple noncovalent interactions between the C[3]A and guests (Fig. 3b). For example, in the structure of G1@C[3]A, the cyano group on the G1 has close contacts with the methyl group and aromatic ring of C[3]A by hydrogen bond (C – H⋯N) interaction (green dash line A, with the distance of 2.57 Å) and π-π stacking interaction (orange dash line B, with the distance of 3.39 Å). Meanwhile, multiple C – H⋯π interactions (red dash lines C, D, E and F, with the distances of 2.69 Å, 2.82 Å, 2.86 Å and 2.72 Å, respectively) between the hydrogen atoms on the pyridine ring of G1 and the aromatic inner surface of C[3]A can also be found. Independent gradient model (IGM) analysis (Fig. 3c) provides a visual understanding of these noncovalent bonding interactions, which is consistent with SCXRD analysis results. In addition, adjacent host–guest complexes in G1@C[3]A are further stabilized by C – H⋯N hydrogen bond and C – H⋯π interactions (supplementary Fig. 11). With the cooperation of these noncovalent interactions, the host–guest complexes formed by C[3]A and G1 are fixed in a ordered molecular packing network, making the cocrystal mechanically robust. Similarly, multiple C – H⋯π, hydrogen bond and π⋯π interactions also are discovered in the other cocrystals (Fig. 3b, c and supplementary Figs. 12–26). The synergistic effect of these noncovalent interactions plays a critical role in directing the final cocrystal packing structure and morphology, and hence defining their photophysical properties.

## Photophysical properties

To investigate photophysical properties, the solid powders were obtained by manual grinding of as-prepared cocrystals. No obvious structural changes were found by the comparison of experimental powder X-ray diffraction (PXRD) patterns with the calculated lines from the single-crystal data (Supplementary Fig. 27). Moreover, the thermogravimetric analysis (TGA) and differential scanning calorimetry (DSC) studies indicated structural stability of these host–guest cocrystals (Supplementary Figs. 28, 29). Then, the photophysical properties of these solid powders were recorded and the results were summarized in Table 1 and Supplementary Table 9. The ultraviolet-visible (UV-Vis) absorption spectra reveal that all cocrystals exhibit intense absorption bands around 288 nm and weak absorption bands in the region of 350 - 550 nm (Fig. 4a). The former is mainly attributed to the π-π* transition, while the latter can be assigned to the inter-molecular CT transition between the C[3]A and guests, which are significantly different from the absorption spectra of C[3]A and the guests (G1 - G7) (Supplementary Fig. 30). With increasing electron-

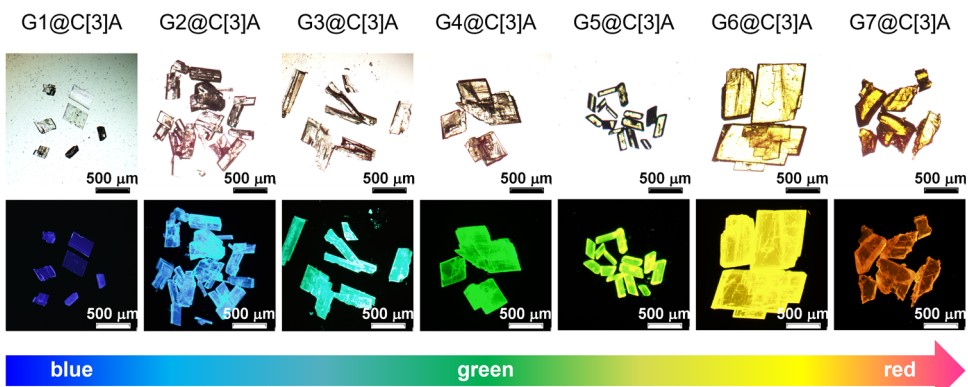

**Fig. 2 | Optical microscopy images of host–guest cocrystals.** Optical microscopy images of G1@C[3]A - G7@C[3]A under sunlight (up) and UV light (down), which display blocky morphology and continuously changing fluorescence from blue to green to red. Scale bar: 500 μm.

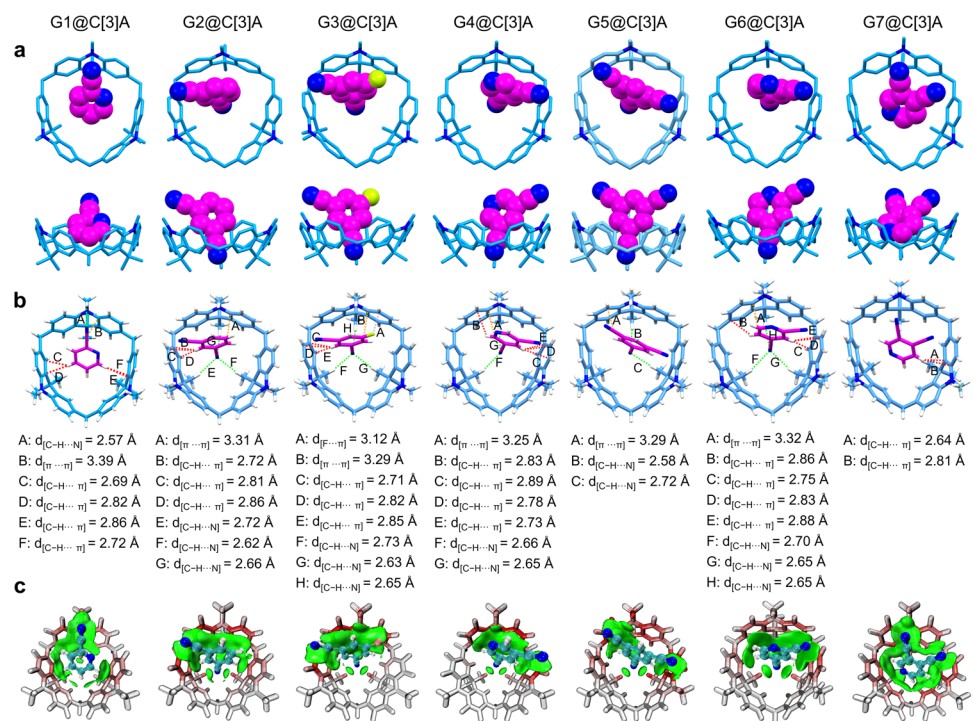

**Fig. 3 | Structural analysis of host−guest cocrystals. a** Crystal structures of G1@C[3]A - G7@C[3]A demonstrating the formation of host−guest inclusion complexes. **b** Illustration of C − H···N, π···π, C − H···π and F···π interactions in the G1@C[3]A - G7@C[3]A, which are marked by green, orange and red dashed lines, respectively. **c** IGM analyses for G1@C[3]A - G7@C[3]A ($\delta g_{inter}$ = 0.004), revealing the multiple noncovalent interactions between the C[3]A and G1 - G7.

### Table 1 | Photophysical data of G1@C[3]A ~ G7@C[3]A

| Entry | G1@C[3]A | G2@C[3]A | G3@C[3]A | G4@C[3]A | G5@C[3]A | G6@C[3]A | G7@C[3]A |
|---|---|---|---|---|---|---|---|
| $\lambda_{abs}{}^a$ [nm] | 288/374 | 288/380 | 288/383 | 288/395 | 288/409 | 288/413 | 288/425 |
| $\lambda_{em}{}^b$ [nm] | 440 | 474 | 493 | 513 | 540 | 570 | 610 |
| $\Phi_{PL}{}^c$ [%] | 24 | 87 | 62 | 65 | 45 | 20 | 14 |
| $\Phi_F{}^d$ [%] | 21 | 49 | 48 | 9 | 12 | 15 | 12 |
| $\Phi_{TADF}{}^e$ [%] | 3 | 38 | 14 | 56 | 33 | 5 | 2 |
| $\tau_F{}^f$ [ns] | 95 | 286 | 300 | 470 | 92 | 211 | 44 |
| $\tau_{TADF}{}^g$ [μs] | 0.6 | 5.3 | 4.9 | 2.6 | 1.8 | 4.0 | 0.9 |
| $E_S/E_T{}^h$ [eV] | 3.117/3.098 | 2.892/2.888 | 2.817/2.812 | 2.730/2.723 | 2.570/2.568 | 2.463/2.460 | 2.309/2.296 |
| $\Delta E_{ST}{}^i$ [meV] | 19 | 4 | 5 | 7 | 2 | 3 | 13 |
| $E_{g\ opt}{}^j$ [eV] | 3.03 | 2.86 | 2.73 | 2.66 | 2.57 | 2.49 | 2.31 |

[a]Absorption peak wavelength.

[b]PL emission maximum.

[c]Total PLQY under vacuum at room temperature evaluated using an integrating sphere.

[d]Fractional PLQY for prompt fluorescence calculated from transient PL experiments and total PLQY measurements.

[e]Fractional PLQY for delayed fluorescence calculated from transient PL experiments and total PLQY measurements.

[f]Prompt fluorescence lifetime obtained from transient PL experiments.

[g]Delayed fluorescence lifetime obtained from transient PL experiments.

[h]$S_1$ and $T_1$ energy levels estimated from the onsets fluorescence at room temperature and phosphorescence spectra at 77 K.

[i]Singlet-triplet energy gap ($\Delta E_{ST}$) calculated from $E_S$ and $E_T$.

[j]Optical gap from Tauc plot.

withdrawing ability of the guests, the CT absorption band moves to longer wavelength from G1@C[3]A to G7@C[3]A (Table 1). The same trend is also observed in the steady state photoluminescence (PL) spectra that the emission wavelengths of the cocrystals are remarkably red-shifted compared to those of the individual components (Fig. 4b). This is due to CT state formation between the C[3]A and guests after photoexcitation. Additionally, these host−guest cocrystals display gradually red-shifted emissions with the emission maximum varying from 440 to 610 nm from G1@C[3]A to G7@C[3]A. Notably, the G1@C[3]A, G4@C[3]A, G7@C[3]A exhibit blue ($\lambda_{em}$ = 440 nm), green

($\lambda_{em}$ = 513 nm) and red ($\lambda_{em}$ = 610 nm) emission, respectively, which represents a type of three primary-color (RGB) emitting systems. By performing Tauc plots on the UV-Vis absorption spectra, the optical band gaps ($E_{g\ opt}$) of all cocrystals can be calculated (Supplementary Fig. 32). The results disclose that the $E_{g\ opt}$ values from G1@C[3]A to G7@C[3]A gradually decrease from 3.03 eV to 2.31 eV, which is very consistent with the red-shifted trend of UV-vis adsorption and steady-state PL spectra. The corresponding Commission Internationale de l'Eclairage (CIE) coordinates of their emissions are calculated (Fig. 4c), which cover the whole visible region from blue to green to yellow and

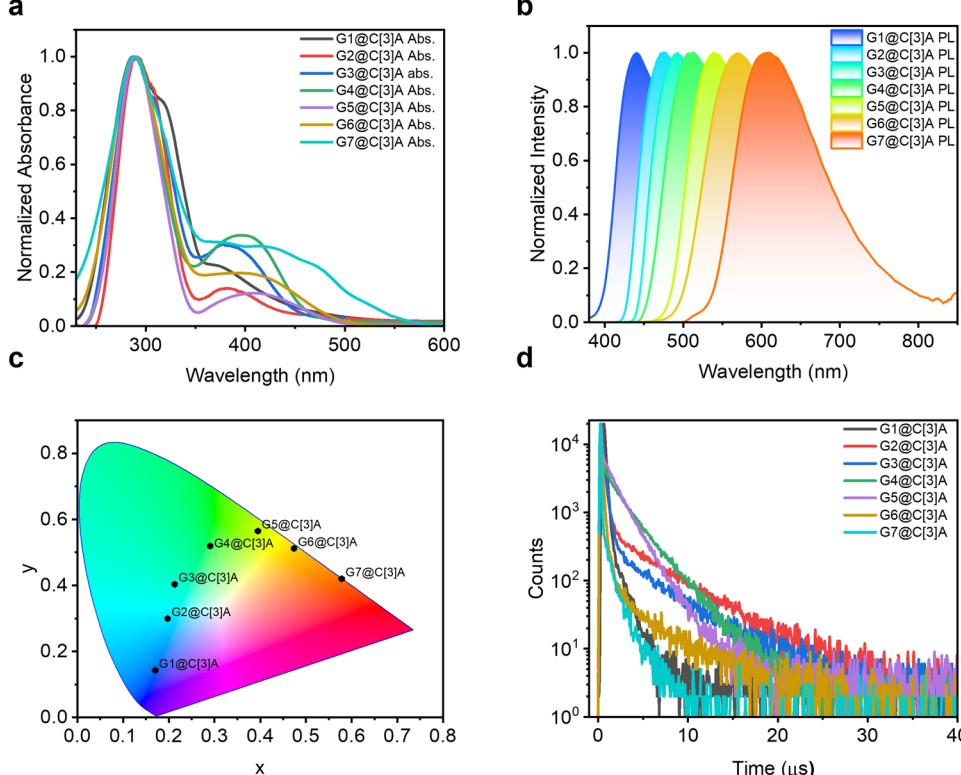

**Fig. 4 | The photophysical investigations of cocrystals. a** UV-Vis absorption (Abs.) and (**b**) steady-state photoluminescence (PL) spectra. **c** Calculated PL emission color coordinates in the CIE 1931 chromaticity diagram. **d** Transient PL decay curves under $N_2$ atmosphere at ambient temperature. All the measurements were carried out in the solid state.

then to red. This is quite interesting phenomenon that enables us to simply achieve full-color-tunable emission by changing the electron-withdrawing ability of the guests.

To investigate the TADF features of these cocrystals, transient PL decay curves of G1@C[3]A - G7@C[3]A were measured. As shown in Fig. 4d, the PL decay profiles of these cocrystals exhibit nanosecond-scale prompt component and microsecond-scale delayed component under $N_2$ atmosphere at ambient temperature, which can be fitted with a biexponential model. In contrast, the transient PL decay curves of individual host and independent guests coincide with the IRF of the instrument, distinguishing them from the long lifetime of these cocrystals (Supplementary Fig. 31). The lifetimes of prompt component ($\tau_F$) for G1@C[3]A - G7@C[3]A are in the range of 44 - 470 ns, while the delayed components has lifetimes ($\tau_{TADF}$) in the range of 0.6 - 5.3 $\mu$s. In addition, temperature-dependent properties of the transient PL spectra of G1@C[3]A - G7@C[3]A also were measured in the temperature range of 77 - 300 K (Supplementary Fig. 33). Apparently, the delayed emission for all cocrystals intensify with an increase in temperature, demonstrating that the RISC processes are enhanced by the thermal energy (Supplementary Fig. 34). To confirm that delayed component fluorescence decays from the same channel as prompt fluorescence, the time-resolved PL spectra of G1@C[3]A - G7@C[3]A were measured (Supplementary Fig. 35). It is found that the spectra positions and profiles under different time delays are highly overlapped with the steady-state PL spectra, indicating that the delayed component fluorescence decay has same channel ($S_1$ – $S_0$) as the prompt fluorescence decay. The influence of oxygen on TADF materials is self-evident, and we also investigated the PL spectra intensity in vacuum and air under the same testing conditions. As shown in Supplementary Fig. 36, the emission intensity of PL spectra under vacuum conditions shows a significant increase compared to

that under air conditions. These experimental results strongly suggest the TADF properties of G1@C[3]A - G7@C[3]A.

To determine $\Delta E_{ST}$ values, low-temperature phosphorescence spectra of the cocrystals were measured at 77 K (Supplementary Fig. 37). The onset of fluorescence spectra at room temperature and phosphorescence spectra were used as the singlet and triplet energy levels, respectively. Then the $\Delta E_{ST}$ values of G1@C[3]A - G7@C[3]A were calculated to be 19, 4, 5, 7, 2, 3, 13 meV, respectively, which were much smaller than normally required 300 meV for TADF materials (Table 1). The PLQYs ($\Phi_{PL}$) of the cocrystals measured by integrating sphere under deoxygenated conditions are found to be 24, 87, 62, 65, 45, 20, 14% for the cocrystals from G1@C[3]A to G7@C[3]A, respectively (Supplementary Figs. 38–44). It is worth noting that these values are quite high relative to reported intermolecular D – A type TADF materials, and the value 87% represents the highest record for such systems. Among the cocrystals, G4@C[3]A with a quantum efficiency ($\Phi_{TADF}$) of 56% for the delayed component accompanied by a RISC rate constants ($k_{RISC}$) of $1.4 \times 10^5 \, s^{-1}$ shows the largest ratio of delayed component (86%) relative to the overall fluorescence, implying its more efficient RISC process than the rest of cocrystals. In addition, a larger $k_{RISC}$ is more conducive to triplet exciton utilization, while reducing triplet-triplet annihilation (TTA) and singlet-triplet annihilation (SAT), which is critical for the efficiency roll-off of smaller materials in devices. The reason for such high PLQYs should be ascribed to high-efficiency electronic communication mediated by host–guest interactions. On one hand, the donor and acceptor are to some extent fixed in the solid state and CT excited state can be more easily formed and stabilized. On the other hand, the abundant noncovalent interactions in the cocrystals provide a rigid environment to slow down the molecular rotation and vibration, and therefore suppress the non-radiative decay for efficient TADF[53,54]. The fast rate of radiation decay ($k_F$) allows for the timely use of excitons from the $T_1$ reverse

intersystem crossing to $S_1$ to obtain more efficient luminescence. For this reason, these cocrystals have fast radiative decay rates, which are estimated to be approximately $10^5\,s^{-1}$ or even higher. Moreover, these cocrystals all have the pretty high $k_F$ values, and they can effectively avoid the $S_1$ state excitons obtained by RISC to return to $T_1$ state through intersystem crossing (ISC), which is essential for obtaining high PLQYs (Supplementary Table 9).

## Theoretical calculations

To gain an in-depth understanding of the effect of variation of the guests on the TADF properties of the supramolecular materials, quantum-chemical calculations were performed. The structural optimization of G1@C[3]A - G7@C[3]A were conducted by density functional theory (DFT) calculations with the structures obtained from X-ray single crystal diffraction data. Optimized molecular geometries, energy levels of the HOMO and LUMO and the respective frontier orbital distributions are presented in Fig. 5. The excited $S_1$ and $T_1$ states were computed using the above-optimized structures with time-dependent DFT (TD-DFT) calculations. In the optimized ground-state structures, all the guests are exactly located in the cavity of C[3]A to form host–guest complexes, which are well matched with the results of SCXRD analysis. The energy levels of HOMO of G1@C[3]A - G7@C[3]A are predominantly located on the C[3]A. Whereas, the LUMO orbitals are exclusively concentrated on the guests, indicating the electron transfer from the C[3]A to guests. Based on the results of TD-DFT calculations, we carried out the hole-electron analysis on the $S_1$ states via the Multiwfn program to investigate the distribution of holes and electrons during the $S_0 - S_1$ excitation process[55]. It is found that the holes are mainly distributed on the C[3]A in all cases, with the electrons centred on the guests (Supplementary Fig. 45). In addition, all cases display distance–dependent character that the larger hole-electron separation distance (D_idx), smaller hole-electron overlap integral index (Sr) and positive CT characteristic index (t_idx) (Supplementary Table 10). From the calculated $S_1$ and $T_1$ energy levels, we estimate the $\Delta E_{ST}$ for G1@C[3]A - G7@C[3]A to be 28, 2, 2, 10, 8, 4 and 9 meV, respectively, which are in good agreement with the trends observed experimentally.

The above TD-DFT calculations show that the $S_1$ excited states of G1@C[3]A - G7@C[3]A are dominated by the charge transfer processes form the C[3]A to the guests. Therefore, the emission wavelengths of these cocrystals greatly depend on the HOMO energy levels of C[3]A as well as LUMO energy levels of guests. As expected, these cocrystals have similar HOMO energy levels in the range of −4.98 to −5.14 eV. Whereas, their LUMO energy levels are very different due to the influence of electron-withdrawing ability of the guests. With the increase of electron-withdrawing ability, the LUMO energy levels tend to decrease. The corresponding HOMO-LUMO energy gaps ($E_g$) for all cocrystals also were estimated according to the theoretical results, which can be modulated systematically from 3.98 eV (G1@C[3]A) to 2.99 eV (G7@C[3]A). This trend suggests that the CT strengths have a big impact on the photophysical properties of TADF materials. It is rational and feasible that the emission colors of the supramolecular TADF materials presented in this work can be readily tuned by using different electron-withdrawing guests.

## Discussion

In conclusion, we have presented a general and effective supramolecular strategy for the fabrication of color-tunable TADF materials. With this strategy, we have prepared a series of host–guest cocrystals with the pronounced CT character by adopting calix[3]acridan as electron donor and various commercially available cyano-substituted aromatics as electron acceptor. All these cocrystals possess very small $\Delta E_{ST}$ that originates from naturally separated HOMO and LUMO levels, with subsequent efficient TADF emission. Their emission colors can also be continuously changed from blue to red with the emission wavelengths changing over a 170 nm range, which cover the entire whole visible region. The experimental and theoretical studies revealed that the tunable emission was essentially due to elaborated controlled charge transfer strength between the host and guests and hence regulable band gaps. In other words, the TADF properties of this kind of supramolecular materials can be tailored by altering the guests and the formation of host–guest cocrystals. Compared to intramolecular D – A type TADF materials, the fabrication of supramolecular TADF materials is facile and avoids any tedious covalent synthesis. Notably, this supramolecular TADF materials can achieve a high PLQY up to 87%, which is much higher than previously reported intermolecular D – A type TADF materials. This work here not only provides a universal and promising approach for the fabrication of full-color-tunable TADF materials, but also offers a solution to breaking the limitation of both small $\Delta E_{ST}$ and high PLQY for intermolecular D – A type TADF materials. In our following research, we will extend this strategy to the fabrication of more advanced supramolecular TADF materials including polymers and nanoparticles, and find their applications in electroluminescence, organic lasers, biological field and so on.

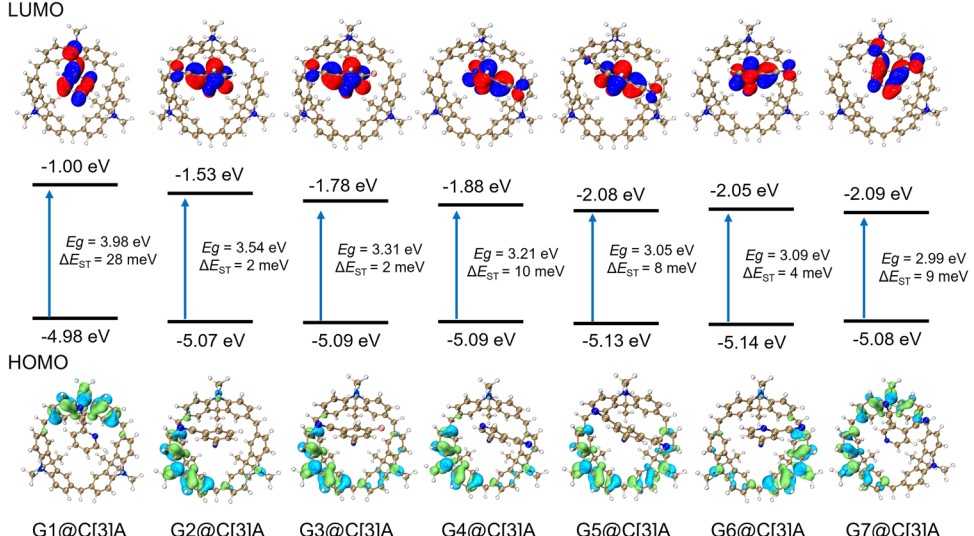

**Fig. 5 | Theoretical calculations.** Frontier-orbital (HOMO/LUMO) distributions, energy levels and energy gaps ($\Delta E_{ST}$) between the $S_1$ and $T_1$ for G1@C[3]A - G7@C[3]A characterized by DFT and TD-DFT calculations.

## Methods

### Materials

All chemicals and solvents were purchased from commercial suppliers and used without further purification. Calix[3]acridan (C[3]A) was synthesized and purified using previously reported procedures, details of which are given in the Supplementary information[40]. Anhydrous dichloromethane was dried with 4 Å molecular sieves. Flash column chromatography was performed on 200–300 mesh silica gel. 9,9-Mimethyl-9,10-dihydroacridine ($C_{15}H_{15}N$, 98%) was purchased from Heowns. 2-Cyanopyridine ($C_6H_4N_2$, 99%), defined as G1, was purchased from Heowns. 1,3-Dicyanobenzene ($C_8H_4N_2$, 98%) defined as G2, was purchased from Macklin. 3,5-Dicyanofluorobenzene ($C_8H_3FN_2$, 98%), defined as G3, was purchased from Macklin. Pyridine-3,5-dicarbonitrile ($C_7H_3N_3$, 95%), defined as G4, was purchased from Bidephorm. 1,3,5-Benzenetricarbonitile ($C_9H_3N_3$, 98%), defined as G5, was purchased from Innochem. 2,4-Pyridinedicarbonitrile ($C_7H_3N_3$, 97%), defined as G6, was purchased from Aladdin. 3,4-Pyridinedicarbonitrile ($C_7H_3N_3$, 98%), defined as G7, was purchased from Aladdin. All cocrystals were obtained by solvent evaporation or diffusion of the mixture solution of the 1:1 C[3]A and guests.

### $^1$H NMR and $^{13}$C NMR spectroscopy experiments

The NMR spectroscopy experiments of C[3]A were recorded on the Bruker Avance III 400 MHz spectrometer and spectra are shown in Supplementary Figs. 1, 2.

### High resolution mass spectrometry (HRMS)

Electrospray ionization mass spectra (ESI-MS) were recorded on the Thermo Fisher Exactive high-resolution LC-MS spectrometer and spectrum is shown in Supplementary Fig. 3.

### Single crystal X-ray crystallography

Single-crystal X-ray diffraction data were collected on a Bruker Smart APEXII CCD diffractometer using graphite monochromated Cu Kα ($λ = 1.54184$ Å) radiation at 170 K and the detailed experimental parameters are summarized in Supplementary Tables S2–8. Olex 2 and PLATON software were used for the structure refinement.

### Powder X-ray diffraction analysis

Powder X-ray diffraction were carried out on a SmartLab X-ray diffractometer with a total reflection silicon wafer.

### Thermogravimetric analysis

Thermogravimetric analysis were measured on Perkin Elmer® Pyris 1 TGA from 25 °C to 700 °C at 10 °C/min.

### Differential scanning calorimetry

Differential scanning calorimetry were performed on TA Q2000 from 25 °C to 100 °C.

### Optical characterizations

Optical and fluorescence microscopy images were captured with an Olympus IX83 microscope. Optical microscopy pictures were obtained in DIC mode. Fluorescence microscopy images were obtained in the fluorescence mode. Solid-state UV-vis absorption spectra were measured by a reflectance mode on a Perkin Elmer® Lambda 1050+ spectrometer from 200 to 600 nm with $BaSO_4$ as a reference. The photoluminescence (PL) spectra, phosphorescence (Phos.) spectra, transient PL decay characteristics, time-resolved PL spectra, and temperature dependence experiments were measured on an Edinburgh Instruments FLS1000 spectrometer. Xenon lamp as light source for the PL spectra, microsecond lamp as light source for Phos. spectra and time-resolved PL spectra, and VPL laser as light source for transient PL and temperature dependence experiments. Photoluminescent quantum yields (PLQYs) were obtained on a HORIBA FluoroMax

spectrometer with an integrating sphere. The cocrystals for PLQY measurement were sealed in the quartz glass under the nitrogen atmosphere of the glovebox to remove the air.

### Theoretical calculation

All theoretical calculations were performed by the Gaussian 09 package, details of which were given in Supplementary information. The atomic coordinate files obtained after DFT optimization are included as Supplementary Data files 1–14 in the Supplementary Files.

## Data availability

All data supporting this study including detailed methods and experimental details, photophysical properties studies, Powder X-ray diffraction (PXRD) patterns and crystallographic details of all of the crystal structures are available in Manuscript and Supplementary information. In addition, these data are available upon request from the authors. The X-ray crystallographic coordinates for the structures reported in this study have been deposited at the Cambridge Crystallographic Data Centre (CCDC), under deposition numbers 2293458-2293463, 2293490. These data can be obtained free of charge from The Cambridge Crystallographic Data Centre via www.ccdc. cam.ac.uk/data_request/cif.

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

## Acknowledgements

We thank the National Natural Science Foundation of China (22031010, 22371277, 92256304) for the financial support. We also thank Mrs. Tongling Liang for her help in the crystal refinements.

## Author contributions

H.-Y.L. and C.-F.C. conceived this project and designed the experiments. N.X. conducted the experiments. H.-Y.Z. and N.X. wrote the manuscript. Y.H. and M.L. were involved in the discussions and contributed to the manuscript preparation. N.X., H.-Y.Z., M.L., H.-Y.L., and C.-F.C. revised and finalized the manuscript.

## Competing interests

The authors declare no competing interests.
