## [Peer Review File · Nature Communications]

A general supramolecular strategy for fabricating full-color-tunable thermally activated delayed fluorescence materialsREVIEWER COMMENTS

Reviewer #1 (Remarks to the Author):

Chen and Lu et al not only proposed a general supramolecular strategy for fabricating full-color-tunable TADF materials, but also established a reliable supramolecular protocol. Among them, a series of cyano-substituted aromatic acceptors were selected, 2-cyanopyridine (G1), 1,3-dicyanobenzene (G2), 3,5-dicyanofluorobenzene (G3), 3,5-dicyanopyridine (G4), 1,3,5- tricyanobenzene (G5), 2,4-dicyanopyridine (G6) and 3,4-dicyanopyridine (G7), as guest molecules to fabricate TADF materials through the formation of host-guest cocrystals. All these cocrystals possess very small ΔE_{ST} that originates from naturally separated HOMO and LUMO levels, with subsequent efficient TADF emission. Their emission colors can be continuously changed from blue(440 nm) to red(610 nm) with the emission wavelengths changing over a 170 nm range, which cover the entire whole visible region. Meanwhile, a high photo luminescence quantum yield of up to 87% for the supramolecular TADF material was achieved. In this sense, this work can be recommended for publication in Nature Communications after minor revision. .

1.The Fig. 3b is not very clear. Please modify it in revision.

2.The thermogravimetric analysis and differential scanning calorimetry should be provided to evaluate the thermal stability of G1@C[3]A~G7@C[3]A.

3.Can you provide a detailed comparison of the advantages and disadvantages of this supramolecular TADF materials and intermolecular D-A type TADF materials?

4.Some typical references about long-lived luminescence materials are suggested to be cited, such as 10.1016/j.chempr.2023.05.023;10.1002/ange.202203254;10.1002/adma.202204415;10.1021/jacs.2c02076.

Reviewer #2 (Remarks to the Author):

In this manuscript, as their continuous research interest in macrocyclic chemistry and supramolecular luminescent materials, Chen and coworkers demonstrated the development of a facile and feasible strategy to fabricate TADF materials based on the formation of host-guest cocrystals. More importantly, such novel strategy is quite general, thus leading to the continuous modulation of the emission colors from blue to red with high photoluminescence quantum yields. According to this interesting and important work, supramolecular chemistry again shows its great power in the creation of advanced functional materials. Definitely, this research will receive much attention from the community not only from supramolecular chemists but also from the materials science. Thus publication is strongly recommended, only some minor issues as listed below need to be addressed:

1. This type of novel supramolecular TADF materials are fabricated in co-crystal form based on host-guest chemistry, how about their host-guest chemistry and emission behaviors in solution state?

2. For practical use, the quality and scale are crucial. For these TADF cocrystals, how large scale can they

be prepared?

3. Supramolecular strategy can not only be used for non-covalent synthesis, but also important method to achieve stimuli-responsive property. Thus, in this system, is it possible to realized switchable emission regulation through precisely controlling the host-guest complexations?

Reviewer #3 (Remarks to the Author):

The manuscript entitled "A General Supramolecular Strategy for Fabricating Full-Color-Tunable TADF Materials" reports unique and interesting results that could be used as a general design principle for creating co-crystals with tuned energetics and photophysics. The authors claim a quantum yield approaching 87% and long-lived emission lifetimes consistent with TADF characteristics. In principle, it is possible to obtain TADF characteristics via the "through space charge transfer" mechanism. Although this manuscript provides unique insights into co-crystal design for optical tuning, it needs more in-depth solid-state and photophysical analysis before publication, specifically in this journal. It is unclear whether the photophysics reported here are due to TADF characteristics or room temperature phosphorescence. Additionally, including device characterization will greatly improve this manuscript. Specific comments are enclosed below:

- 1) The main contribution of this manuscript is the co-crystal synthetic approach for obtaining unique photophysical properties. An expert in crystallography should evaluate the crystals in detail.
- 2) It would be great if the authors could include the photophysics of the independent guest and host crystals to accentuate the unique photophysics of the co-crystals.
- 3) Authors should discuss their findings within the "through space charge transfer" context.
- 4) I recommend that the authors discuss in greater detail the rate of electronic transitions within the context of Masui et al. (10.1016/j.orgel.2013.07.010) and Vazquez et al. (doi.org/10.1021/jacs.0c01225), Table S9.
- 5) The authors should include an emission spectrum of the crystals at different temperatures. A clear decrease in emission intensity upon decreasing the temperature, consistent with TADF behavior, should be clearly observed. This will unequivocally suggest photophysics governed by TADF characteristics.
- 6) This manuscript would greatly benefit from device characterization as proof of harvesting the triplet state in operational conditions.

Reviewer #4 (Remarks to the Author):

In this work, a general supramolecular strategy for fabricating full-color-tunable TADF materials was proposed. Consequently, a series of host–guest cocrystals were prepared by electron-donating calix[3]acridan and various electron-withdrawing guests. Owing to the intermolecular charge transfer mediated by multiple noncovalent interactions between the host and guests, these cocrystals all displayed efficient TADF properties. Especially, by delicately modulating the electron-withdrawing ability of the guest molecules, the emission colors of these TADF materials could be continuously tuned from blue (440 nm) to red (610 nm). Meanwhile, a high photoluminescence quantum yield of up to 87% for the supramolecular TADF material was achieved. This research not only provides an alternative and general strategy for the fabrication of full-color-tunable TADF materials, but also establishes a reliable supramolecular protocol toward the design of advanced luminescent materials.

1. Author have to explain the choice of PBE05 functional and the 6-31g(d) basis set for the DFT calculations. And B3LYP functional and the 6-311g(d) basis set for TD-DFT.
2. In Supplementary Table 2–8, the space groups are in the wrong format. The "c and n" in "P21/c and P21/n " should be in italics, while the "P" in "P-1" should also be in italics.
3. What does RISC stand for? to be defined in the initial description.
4. On page 15, the descriptions of lines 269 and 267 should be consistent.
5. The author mentions in the manuscript that "this supramolecular TADF materials can achieve a high PLQY up to 87%", how was the 87% obtained?
6. No highlights are provided. Author should standardize the format of all references.

Responses to the reviewers' comments:

We are grateful to the reviewers for the comments and suggestions on our manuscript "A general supramolecular strategy for fabricating full-color-tunable TADF materials" (Manuscript number: NCOMMS-23-48977A-Z), which are very helpful for us to improve our manuscript.

According to the comments and suggestions of reviewers, Manuscript and Supplementary information have been carefully checked and revised. The detailed changes and responses are as follows.

Point by Point Responses to Reviewer 1:

1) *The Fig. 3b is not very clear. Please modify it in revision.*

Answer: According to the suggestion, the Fig. 3 has been redrawn for clarity. Thank you.

2) *The thermogravimetric analysis and differential scanning calorimetry should be provided to evaluate the thermal stability of G1@C[3]A~G7@C[3]A.*

Answer: According to your suggestion, we have studied the thermal stability of these cocrystals by thermogravimetric analysis and differential scanning calorimetry (see Supplementary Fig. 17 and 18). The results showed that these TADF materials exhibited good thermal stability. Since strong acidic HF released by guest decomposition in G3@C[3]A may corrode the crucible of TGA instrument, TGA of G3@C[3]A is not tested.

3) *Can you provide a detailed comparison of the advantages and disadvantages of this supramolecular TADF materials and intermolecular D-A type TADF materials?*

Answer: As mentioned in the manuscript, supramolecular TADF materials have the following advantages compared to intermolecular D-A type TADF materials: (1) the preparation method is simple and also has wide universality; (2) the TADF properties can be tailored; (3) full-color emission can be readily achieved; and (4) more applications besides in OLEDs can be found (such as in the controllable and/or switchable recognition, fluorescence sensor and detection). However, the studies on supramolecular TADF materials are still in the infancy. Especially, the applications of supramolecular TADF materials need to be explored, and they will be one of the major targets in this research area.

4) *Some typical references about long-lived luminescence materials are suggested to be cited, such as 10.1016/j.chempr.2023.05.023; 10.1002/ange.202203254; 10.1002/adma.202204415; 10.1021/jacs.2c02076.*

Answer: Thank you for the suggestion. According to the suggestion, the related typical references have been cited (see Ref. 33, 34, 38, and 44).

Point by Point Responses to Reviewer 2:

1) *This type of novel supramolecular TADF materials are fabricated in co-crystal form based on host-guest chemistry, how about their host-guest chemistry and emission behaviors in solution state?*

Answer: The host-guest interactions between the macrocycle and the guests (neutral molecules) are weak, so we didn't observe the formation of host-guest complexes in solution by ^1H NMR experiments. Correspondingly, no TADF property could be detected in solution.

2) *For practical use, the quality and scale are crucial. For these TADF cocrystals, how large scale can they be prepared ?*

Answer: By means of easy-to-operate cocrystallization method, high-quality TADF cocrystals could be prepared on a several hundred milligram scale only once, which would significantly meet the requirements for the further applications.

3) *Supramolecular strategy can not only be used for non-covalent synthesis, but also important method to achieve stimuli-responsive property. Thus, in this system, is it possible to realized switchable emission regulation through precisely controlling the host-guest complexations?*

Answer: Thank you very much for the comments and suggestions. Owing to the dynamic and reversible nature of host-guest interactions, it is entirely possible to develop stimuli-responsive TADF property of this type of supramolecular materials by external stimuli including temperature, vapor, mechanical forces and so on. Studies along this line are currently underway in our laboratory and will be reported in due course.

Point by Point Responses to Reviewer 3:

1) *It is unclear whether the photophysics reported here are due to TADF characteristics or room temperature phosphorescence.*

Answer: We can confirm that the photophysics reported here are TADF rather than room temperature phosphorescence for the following reasons:

- (1) For cocrystal systems from **G1@C[3]A** to **G7@C[3]A**, the ΔE_{ST} values obtained by fluorescence spectra at 300K and phosphorescent spectra at 77K are 19, 4, 5, 7, 2, 3, 13 meV, respectively (see Manuscript Table 1), which are sufficiently small to ensure effective reverse intersystem crossing (RISC) to obtain TADF properties.
- (2) We conducted fitting analysis on the transient spectra of cocrystal materials from **G1@C[3]A** to **G7@C[3]A** at 300K, and found that the microsecond scale lifetimes are 0.6 μs , 5.3 μs , 4.9 μs , 2.6 μs , 1.8 μs , 4.0 μs , and 0.9 μs , respectively (see Manuscript Table 1). The relatively short lifetime is more in line with TADF rather than room temperature

phosphorescence.

- (3) We also tested the spectra of those cocrystal crystals at 300 K without as well as with 5 μ s, 50 μ s, and even 100 μ s delay, respectively. As shown in Supplementary Fig. 24, compared with fluorescence spectrum without any delay, the difference between the position and profile of the spectra can be ignored in the spectra at different delay times, suggesting that the delay component at 300K is dominated by TADF rather than room temperature phosphorescence.
- (4) To further confirm the TADF properties of these crystalline materials, we analyzed the temperature-dependent transient PL decay curves in detail, which can be divided into three regions (see Supplementary Fig. 23). Take **G2@C[3]A** as an example, the luminescence up to 1.86 μ s comes from prompt fluorescence (PF), whose luminescence intensity is temperature-independent. From 1.86 μ s to around 14 μ s, the emission intensity gradually increases with the increase of temperature, indicating significant TADF properties. For those after 14 μ s gradually reduce with the increase of temperature, which is mainly due to the appearance of phosphorescence in crystals at low temperatures. Based on the existing data analysis, although there is inevitably a small amount of phosphorescence at 300K, TADF still accounts for a larger proportion. Additionally, the temperature-dependent transient PL decay curves with such three regions are widespread in reported TADF materials (such as: *Adv. Mater.* **33**, 2008032 (2021); *Angew Chem. Int. Ed.* **61**, e202209343 (2022); *Angew Chem. Int. Ed.* **58**, 17651–17655 (2019); *Angew Chem. Int. Ed.* **57**, 16407–16411 (2018); *Adv. Mater.* **35**, 2301929 (2023)).
- (5) In our previous work (*Angew. Chem. Int. Ed.* *61*, e202117872 (2022)), we have confirmed that the cocrystal material with macrocycle **C[3]A** and one similar electron acceptor showed significant TADF property. Recently, Chou et al have utilized the same strategy to develop a kind of TADF host-guest complex by blocking the electron donor within the acceptor cage. By using the TADF host-guest complex as emitter, they have fabricated efficient OLED device with EQE_{max} up to 15.2% (*Nat. Chem.* doi:10.1038/s41557-023-01357-0 (2023)). Our series of experimental results and Chou's reports confirm that the host-guest cocrystal materials indeed possess TADF properties.

Based on the above compelling reasons and numerous reported TADF materials, we fully believe that the photophysics reported here are due to TADF, not room temperature phosphorescence.

2) *The main contribution of this manuscript is the co-crystal synthetic approach for obtaining unique photophysical properties. An expert in crystallography should evaluate the crystals in detail.*

Answer: The crystal structure data involved in this study were uploaded to the Cambridge crystallographic database (CCDC), and the crystallographic information file (CIF) passed the Checkcif procedure without level A and B alerts.

3) It would be great if the authors could include the photophysics of the independent guest and host crystals to accentuate the unique photophysics of the co-crystals.

Answer: According to your suggestion, the absorption and emission spectra of including calix[3]acridan and studied guests have been provided (see Supplementary Fig. 19). In comparison, typical CT absorption peaks were found in the absorption spectra of cocrystals, accompanied by red-shifting of the emission peaks. In addition, we tested the transient PL decay curves of independent calix[3]acridan and studied guests, and the results showed that the individual components did not show delayed lifetimes (see Supplementary Fig. 20), which could further demonstrate the unique photophysics of the cocrystals.

4) Authors should discuss their findings within the "through space charge transfer" context.

Answer: As we know, through-space charge transfer (TSCT) excited states have been widely explored for fabricating TADF materials. From the perspective of charge-transfer mechanisms, our systems have similarities with the TSCT-type TADF materials. So, our research work can be discussed as TSCT-type systems. Thank you for your suggestion, we have already mentioned "through-space charge transfer" in our manuscript (see page 6, line 4).

5) I recommend that the authors discuss in greater detail the rate of electronic transitions within the context of Masui et al. (10.1016/j.orgel.2013.07.010) and Vazquez et al. (doi.org/10.1021/jacs.0c01225), Table S9.

Answer: According to your suggestion, we have discussed the rate of electronic transitions in more detail, and they are reflected in the manuscript (see page 15 and 16 in Manuscript).

6) The authors should include an emission spectrum of the crystals at different temperatures. A clear decrease in emission intensity upon decreasing the temperature, consistent with TADF behavior, should be clearly observed. This will unequivocally suggest photophysics governed by TADF characteristics.

Answer: According to your suggestion, we have measured the emission spectra of these crystalline materials from 300K to 77K, but we did not observe the expected results of a clear decrease in emission intensity upon decreasing the temperature. Actually, the spectra at low temperatures are the sum of phosphorescence and TADF intensities. And in most cases, as the temperature decreases, the enhancement of phosphorescence is greater than the weakening of TADF, which might be the reason that a clear decrease of emission intensity could not be usually observed. This phenomenon was also observed in the vast majority of luminescent materials

participated by triplet states (such as: *Angew Chem. Int. Ed.* **61**, e202209343 (2022); *Angew Chem. Int. Ed.* **58**, 17651–17655 (2019); *J. Mater. Chem. C* **10**, 11607–11613 (2022), *Angew Chem. Int. Ed.* **60**, 18059–18064 (2021)). Thus, the temperature-dependent emission spectrum is not a key indicator for verifying the TADF properties of materials.

In our system, we have clearly confirmed the TADF properties of the cocrystal materials by small ΔE_{ST} , short delayed lifetime, and typical temperature-dependent transient PL decay curves as well as delayed emission spectra. To further confirm the TADF properties of the crystalline materials, we have also conducted the temperature-dependent transient PL decay curves with three regions (see Supplementary Fig. 23) in detail, which provided another rigid evidence for the TADF properties of our cocrystal materials. The temperature-dependent transient PL decay curves with such three regions are also widespread in the reported TADF materials (such as: *Adv. Mater.* **33**, 2008032 (2021); *Angew Chem. Int. Ed.* **61**, e202209343 (2022); *Angew Chem. Int. Ed.* **58**, 17651–17655 (2019); *Angew Chem. Int. Ed.* **57**, 16407–16411 (2018); *Adv. Mater.* **35**, 2301929 (2023)).

7) This manuscript would greatly benefit from device characterization as proof of harvesting the triplet state in operational conditions.

Answer: Thank you for the suggestion. In fact, the cocrystal TADF materials are not suitable to fabricate devices by neither spin-coating nor vacuum evaporation methods, which is mainly due to the inability to ensure the crystalline state of material during the device process. Thus, it is still a considerable challenge to incorporate the cocrystal materials into the device characterization. Nevertheless, the crystalline materials in this work display high PLQY, sufficiently small ΔE_{ST} values and microsecond delayed lifetime under photoexcitation conditions, which indicates obvious TADF properties.

Point by Point Responses to Reviewer 4:

1) Author have to explain the choice of PBE05 functional and the 6-31g(d) basis set for the DFT calculations. And B3LYP functional and the 6-311g(d) basis set for TD-DFT.

Answer: In our work, the ground state geometries of all studied cocrystals were optimized by the DFT method with the PBE0 functional and the 6-31g(d) basis set. Accordingly, we should carry out the TD-DFT calculations at the same level. So, we changed the TD-DFT calculation function from B3LYP to PBE0, and the corresponding calculation results in the manuscript have been revised.

2) In Supplementary Table 2–8, the space groups are in the wrong format. The "c and n" in "P21/c and P21/n" should be in italics, while the "P" in "P-1" should also be in italics.

Answer: All these formatting issues in the supplementary information that you mentioned have been handled. Thank you for the helps.

3) What does RISC stand for? to be defined in the initial description.

Answer: RISC is an abbreviation for reverse intersystem crossing, which has been defined in the initial description of the text.

4) On page 15, the descriptions of lines 269 and 267 should be consistent.

Answer: The descriptions about PLQYs on page 15 have been revised.

5) The author mentions in the manuscript that "this supramolecular TADF materials can achieve a high PLQY up to 87%", how was the 87% obtained?

Answer: The PLQYs have been obtained on a HORIBA FluoroMax spectrometer with an integration sphere by using the cocrystals in oxygen-free environments.

6) No highlights are provided. Author should standardize the format of all references.

Answer: According to the suggestion, we have standardized the format of all references.

REVIEWERS' COMMENTS

Reviewer #1 (Remarks to the Author):

The authors have revised the manuscript fully and it is suitable for publication as it is.

Reviewer #2 (Remarks to the Author):

The manuscript was carefully revised, and all of my concerns were well solved, thus I think it can be accepted in the present format.

Reviewer #3 (Remarks to the Author):

The authors addressed my remarks professionally and in detail. I strongly believe that the publication of this manuscript will be of great benefit to the nature readership—very good and cool stuff.

The next big breakthrough could come from making co-crystal devices.

Reviewer #4 (Remarks to the Author):

Authors carefully addressed the comments raised by the reviewer. Therefore, I recommend this revised manuscript for publication in NC.